# Universal relations in flat band superconducting bipartite lattices

G. Bouzerar[*] and M. Thumin

*Université Grenoble Alpes, CNRS, Institut NEEL, F-38042 Grenoble, France*

(Dated: August 17, 2023)

Unconventional flat band (FB) superconductivity, as observed in van der Waals heterostructures, could open promising avenues towards high-$T_c$ materials. Indeed, in FBs, pairings and superfluid weight scale linearly with the interaction parameter, such an unusual behaviour justifies strategies to promote FB engineering. Bipartite lattices (BLs) which naturally host FBs could be particularly interesting candidates. By revealing a hidden symmetry of the quasi-particle eigenstates, we demonstrate that pairings and superfluid weight obey universal relations in BLs. Remarkably, these general properties are insensitive to disorder as long as the bipartite character is protected.

Over the past decade, one witnesses a growing interest for a family of emerging materials: the flat band (FB) systems [1–5]. In FB compounds, the kinetic energy being quenched, the electron-electron interaction energy is the unique relevant energy scale giving access to strongly correlated physics. FBs are found at the origin of an unconventional form of superconductivity (SC) of interband nature. In FBs, superfluid weight (SFW) and critical temperature scale linearly with the effective interaction amplitude $|U|$ [6–8] which contrasts with the dramatic $e^{-1/(\rho(E_F)|U|)}$ scaling in standard BCS theory. Recently, it has been shown that the Bogoliubov de Gennes (BdG) approach is astonishingly quantitatively accurate in describing SC in FBs. Surprisingly, the agreement revealed between exact methods and BdG concerns systems where quantum fluctuations are the strongest: one-dimensional systems such as the sawtooth chain, the Creutz ladder and other FB system. The SFW obtained with BdG and that calculated with density matrix renormalization group (DMRG) were found to agree impressively [9].

In the framework of the attractive Hubbard model (AHM) in bipartite lattices (BLs) and within BdG theory, we demonstrate universal sum-rules and other relations that pairings and SFW obey in half-filled and partially filled FB systems. Some of the relations proved here in a general context have been established for peculiar two dimensional BLs assuming uniform pairings in each sublattice [8, 10]. For the sake of concreteness, all the properties that will be proved are illustrated in the Supplemental Material [11] on a typical two dimensional BL. Finally, in this work, we restrict ourself to $T = 0$.

BLs consist in two sublattices $\mathcal{A}$ and $\mathcal{B}$, with respectively $\Lambda_A$ and $\Lambda_B$ orbitals per cell where $\mathcal{A} = \{A_1, A_2, ..., A_{\Lambda_A}\}$ and $\mathcal{B} = \{B_1, B_2, ..., B_{\Lambda_B}\}$. In the absence of interaction, the spectrum consists exactly in $N_{fb} = \Lambda_B - \Lambda_A$ FBs located at $E = 0$ and $2\Lambda_A$ dispersives bands (DBs) being symmetric because of chiral symmetry. The total number of orbitals per cell is defined as $\Lambda = \Lambda_B + \Lambda_A$. Here, $\Lambda_B > \Lambda_A$ is assumed. The AHM reads,

$$\hat{H} = \sum_{Ii,Jj,\sigma} t^{IJ}_{A_i B_j} c^{\dagger}_{IA_i,\sigma} c_{JB_j,\sigma}$$
$$-|U| \sum_{I,l,\lambda=A,B} \hat{n}_{I\lambda_l,\uparrow} \hat{n}_{I\lambda_l,\downarrow} - \mu \hat{N}, \qquad (1)$$

$I$, $J$ are cell indices, $A_i$ (resp. $B_j$) labels the orbital in $\mathcal{A}$ (resp. $\mathcal{B}$). $t^{IJ}_{A_i B_j}$ are the hoppings between orbitals in $\mathcal{A}$ and $\mathcal{B}$ only. $c^{\dagger}_{I\lambda_l,\sigma}$ creates an electron of spin $\sigma$, in the orbital $\lambda_l$ of the $I$-th cell and $\hat{n}_{I\lambda_l,\sigma} = c^{\dagger}_{I\lambda_l,\sigma} c_{I\lambda_l,\sigma}$ with $\lambda = A, B$. $|U|$ is the strength of the electron-electron interaction and $\mu$ is the chemical potential.

Within BdG theory, pairings and occupations are calculated self-consistently considering a paramagnetic ground-state, $\langle \hat{n}_{\lambda,\uparrow} \rangle_0 = \langle \hat{n}_{\lambda,\downarrow} \rangle_0 = \frac{1}{2} n_\lambda$, where $\langle \ldots \rangle_0$ means thermal average. The BdG Hamiltonian reads,

$$\hat{H}_{BdG} = \sum_{\mathbf{k}} \begin{bmatrix} \hat{\mathbf{C}}^{\dagger}_{\mathbf{k}\uparrow} & \hat{\mathbf{C}}_{-\mathbf{k}\downarrow} \end{bmatrix} \begin{bmatrix} \hat{h}^{\uparrow}_{\mathbf{k}} & \hat{\Delta} \\ \hat{\Delta}^{\dagger} & -\hat{h}^{\downarrow *}_{-\mathbf{k}} \end{bmatrix} \begin{bmatrix} \hat{\mathbf{C}}_{\mathbf{k}\uparrow} \\ \hat{\mathbf{C}}^{\dagger}_{-\mathbf{k}\downarrow} \end{bmatrix}, \qquad (2)$$

the $(\Lambda)$-dimensional spinor $\hat{\mathbf{C}}^{\dagger}_{\mathbf{k}\sigma} = \left( \hat{\mathbf{C}}^{A\dagger}_{\mathbf{k}\sigma}, \hat{\mathbf{C}}^{B\dagger}_{\mathbf{k}\sigma} \right)^t$ with $\hat{\mathbf{C}}^{\lambda\dagger}_{\mathbf{k}\sigma} = (\hat{c}^{\dagger}_{\mathbf{k}\lambda_1,\sigma}, \hat{c}^{\dagger}_{\mathbf{k}\lambda_2,\sigma}, ..., \hat{c}^{\dagger}_{\mathbf{k}\lambda_{\Lambda_\lambda},\sigma})^t$ where $\lambda = A, B$. Finally, $c^{\dagger}_{\mathbf{k}\lambda,\sigma}$ is the Fourier transform (FT) of $c^{\dagger}_{i\lambda,\sigma}$ and,

$$\hat{h}^{\sigma}_{\mathbf{k}} = \begin{bmatrix} -\mu \hat{\mathbb{1}}_{\Lambda_A} - \hat{V}^A & \hat{h}_{AB} \\ \hat{h}^{\dagger}_{AB} & -\mu \hat{\mathbb{1}}_{\Lambda_B} - \hat{V}^B \end{bmatrix}. \qquad (3)$$

$\hat{h}_{AB}$ is the FT of the tight-binding term in Eq. (1), $\hat{V}^\lambda = \frac{|U|}{2} \text{diag}(n_{\lambda_1}, n_{\lambda_2}, ..., n_{\lambda_{\Lambda_\lambda}})$, with $\lambda = A, B$ and the pairing matrix is,

$$\hat{\Delta} = \begin{bmatrix} \hat{\Delta}^A & 0_{\Lambda_A \times \Lambda_B} \\ 0_{\Lambda_B \times \Lambda_A} & \hat{\Delta}^B \end{bmatrix}, \qquad (4)$$

$\hat{\Delta}^\lambda = \text{diag}(\Delta^\lambda_1, \Delta^\lambda_2, ..., \Delta^\lambda_{\Lambda_\lambda})$, where $\Delta^\lambda_l = -|U| \langle \hat{c}_{I\lambda_l,\downarrow} \hat{c}_{I\lambda_l,\uparrow} \rangle_0$ with $\lambda = A, B$. At half filling ($\mu = -|U|/2$), the density being uniform [12] the diagonal blocks in Eq.(3) vanish. Here, we assume time reversal symmetry, hence pairings can be taken real. In what follows, for any $|U|$, the pairings are real and

positive.

*Hidden symmetry in the BdG eigenstates.-* Let us define positive (respectively negative) eigenstates those with positive (respectively negative) energy. Consider $|\Psi\rangle = (|\mathbf{u}\rangle, |\mathbf{v}\rangle)^t$ an eigenstate of energy $E$, where $|\mathbf{u}\rangle = (|\mathbf{a}\rangle, |\mathbf{b}\rangle)^t$ and $|\mathbf{v}\rangle = (|\bar{\mathbf{a}}\rangle, |\bar{\mathbf{b}}\rangle)^t$. $|\mathbf{a}\rangle$ and $|\bar{\mathbf{a}}\rangle$ (respectively $|\mathbf{b}\rangle$ and $|\bar{\mathbf{b}}\rangle$) are column of length $\Lambda_A$ (respectively $\Lambda_B$).

**Lemma 1**: *Positive (respectively negative) eigenstates can be split into two subsets $\mathcal{S}_+$ and $\mathcal{S}_-$, where, $|\Psi\rangle \in \mathcal{S}_+ \Leftrightarrow |\mathbf{v}\rangle = (|\mathbf{a}\rangle, -|\mathbf{b}\rangle)^t$, and $|\Psi\rangle \in \mathcal{S}_- \Leftrightarrow |\mathbf{v}\rangle = (-|\mathbf{a}\rangle, |\mathbf{b}\rangle)^t$.*
Proof: At half-filling $\hat{H}_{BdG}$ is invariant under particle-hole (PH) transformation which reads,

$$\begin{bmatrix} \hat{\mathbf{C}}^\dagger_{A\uparrow} \\ \hat{\mathbf{C}}^\dagger_{B\downarrow} \end{bmatrix} \overset{PH}{\Longrightarrow} \begin{bmatrix} \hat{\mathbf{C}}_{A\downarrow} \\ -\hat{\mathbf{C}}_{B\uparrow} \end{bmatrix}. \tag{5}$$

Hence, $|\Psi\rangle = (|\mathbf{a}\rangle, |\mathbf{b}\rangle, |\bar{\mathbf{a}}\rangle, |\bar{\mathbf{b}}\rangle)^t \overset{PH}{\Rightarrow} |\Psi_1\rangle = (|\bar{\mathbf{a}}\rangle, -|\bar{\mathbf{b}}\rangle, |\mathbf{a}\rangle, -|\mathbf{b}\rangle)^t$. PH symmetry implies $|\Psi_1\rangle = e^{i\varphi}|\Psi\rangle$, leading to $e^{i\varphi} = \pm 1$. Thus, we are left with two possibilities: (1) $|\Psi\rangle \in \mathcal{S}_+$ or (2) $|\Psi\rangle \in \mathcal{S}_-$ corresponding respectively to $\varphi = 0$ and $\varphi = \pi$. Notice that, if $|\Psi_+\rangle \in \mathcal{S}_+$ has energy $E$, then the eigenstate $\hat{U}|\Psi_+\rangle \in \mathcal{S}_-$ has energy $-E$, since $\hat{U}\hat{H}_{BdG}\hat{U}^\dagger = -\hat{H}_{BdG}$ where $\hat{U} = \begin{bmatrix} 0 & \hat{\mathbb{1}}_\Lambda \\ -\hat{\mathbb{1}}_\Lambda & 0 \end{bmatrix}$.

We proceed further and demonstrate a second lemma that is crucial for what follows.

**Lemma 2**: *For any $|U| \neq 0$, $\mathcal{S}_-$ (respectively $\mathcal{S}_+$) consists exactly in $\Lambda_B$ (respectively $\Lambda_A$) eigenstates of positive or zero energy and $\Lambda_A$ (respectively $\Lambda_B$) eigenstates of strictly negative energy.*
Proof: For what follows, for a given square matrix $\hat{M}$, we define $In(\hat{M}) = (n_m, n_p)$ where $n_m$ is the number of strictly negative eigenvalues and $n_p$ that of the positive or zero eigenvalues. Now, consider $|\phi_n^s\rangle = (|u_n^s\rangle, |v_n^s\rangle)^t$ a QP eigenstate of energy $E_n^s$ in $\mathcal{S}_s$ ($s = \pm$), using Eq.(2) one finds,

$$\hat{\mathcal{H}}^s|u_n^s\rangle = E_n^s|u_n^s\rangle, \tag{6}$$

where the $\Lambda \times \Lambda$ matrices are,

$$\hat{\mathcal{H}}^+ = \begin{bmatrix} \hat{\Delta}^A & \hat{h}_{AB} \\ \hat{h}^\dagger_{AB} & -\hat{\Delta}^B \end{bmatrix}, \hat{\mathcal{H}}^- = \begin{bmatrix} -\hat{\Delta}^A & \hat{h}_{AB} \\ \hat{h}^\dagger_{AB} & \hat{\Delta}^B \end{bmatrix}. \tag{7}$$

For infinitesimal $|U|$, apply a degenerate perturbation theory to the $N_{fb}$ FB eigenstates of $\hat{\mathcal{H}}^\pm|_{|U|=0}$ which have weight on $\mathcal{B}$ orbitals only. The projection of $\hat{\Delta}^B$ in the FB eigenspace being positive definite, it implies that the energy shift of each FB eigenstates of $\hat{\mathcal{H}}^+$ is strictly negative, and strictly positive for those of $\hat{\mathcal{H}}^-$. In other words, it means that $In(\hat{\mathcal{H}}^-) = (\Lambda_A, \Lambda_B)$ and

$In(\hat{\mathcal{H}}^+) = (\Lambda_B, \Lambda_A)$.
Now, assume, there exist a peculiar value $|U_c|$ such that for $|U| < |U_c|$, $In(\hat{\mathcal{H}}^-) = (\Lambda_A, \Lambda_B)$ and $In(\hat{\mathcal{H}}^+) = (\Lambda_B, \Lambda_A)$, and for $|U| > |U_c|$, $In(\hat{\mathcal{H}}^+) = (\Lambda_B - 1, \Lambda_A + 1)$ and $In(\hat{\mathcal{H}}^-) = (\Lambda_A + 1, \Lambda_B - 1)$. At $|U_c|$, $\hat{\mathcal{H}}^-$ and $\hat{\mathcal{H}}^+$ have at least an eigenstate with zero energy, $|u_0^s\rangle = (|\mathbf{a}_0^s\rangle, |\mathbf{b}_0^s\rangle)^t$, $s = \pm$. From Eq.(7) and for $s = +$,

$$(\hat{\Delta}^B + \hat{h}^\dagger_{AB}(\hat{\Delta}^A)^{-1}\hat{h}_{AB})|\mathbf{b}_0^+\rangle = 0,$$
$$|\mathbf{a}_0^+\rangle = -(\hat{\Delta}^A)^{-1}\hat{h}_{AB}|\mathbf{b}_0^+\rangle. \tag{8}$$

$\Delta_i^A > 0$ has been used. $\hat{\Delta}^B + \hat{h}^\dagger_{AB}(\hat{\Delta}^A)^{-1}\hat{h}_{AB}$ is the sum of a positive definite matrix and positive semi definite one, their sum is positive definite and hence zero cannot be an eigenvalue, $|\mathbf{b}_0^+\rangle = [\mathbf{0}]_{\Lambda_B}$ and $|\mathbf{a}_0^+\rangle = [\mathbf{0}]_{\Lambda_A}$ where $|\mathbf{0}]_N$ is the column vector with N zeros. The same proof applies for $|u_0^-\rangle$. This proves the second lemma.

*Pairing sum rule in half-filled bipartite lattices.-* We focus on the negative eigenstates of $\hat{H}_{BdG}$. We define $|\psi_{n+}^<\rangle$ where $n = 1, ...., \Lambda_B$ the normalized eigenstates in $\mathcal{S}_+$ and similarly $|\psi_{m-}^<\rangle$ where $m = 1, ...., \Lambda_A$ those in $\mathcal{S}_-$. We write $|\psi_{n+}^<\rangle = (|u_n^+\rangle, |v_n^+\rangle)^t$ and $|\psi_{m-}^<\rangle = (|u_m^-\rangle, |v_m^-\rangle)^t$. At $T = 0$, pairings are given by,

$$\Delta_l^\lambda = -\frac{|U|}{N_c}\Big(\sum_{\mathbf{k},s=n+}\langle\psi_s^<|\hat{O}_{\lambda_l}|\psi_s^<\rangle + \sum_{\mathbf{k},s=m-}\langle\psi_s^<|\hat{O}_{\lambda_l}|\psi_s^<\rangle\Big), \tag{9}$$

where $\hat{O}_{\lambda_l} = \hat{c}_{-\mathbf{k}\lambda_l,\downarrow}\hat{c}_{\mathbf{k}\lambda_l,\uparrow}$, $\lambda = A, B$ and $l = 1, ..., \Lambda_\lambda$, $n$ runs over $1, ..., \Lambda_B$, and $m$ over $1, ..., \Lambda_A$, $N_c$ being the number of cells. Eq.(9) leads to,

$$\Delta_i^A = -\frac{|U|}{N_c}\Big(\sum_{\mathbf{k},n=1}^{n=\Lambda_B}|a_{ni}^+|^2 - \sum_{\mathbf{k},m=1}^{m=\Lambda_A}|a_{mi}^-|^2\Big),$$
$$\Delta_j^B = \frac{|U|}{N_c}\Big(\sum_{\mathbf{k},n=1}^{n=\Lambda_B}|b_{nj}^+|^2 - \sum_{\mathbf{k},m=1}^{m=\Lambda_A}|b_{mj}^-|^2\Big). \tag{10}$$

The eigenstates beeing normalized, one finally finds the sum-rule,

$$\sum_{j=1}^{\Lambda_B}\Delta_j^B - \sum_{i=1}^{\Lambda_A}\Delta_i^A = \frac{|U|}{2}(\Lambda_B - \Lambda_A). \tag{11}$$

A similar expression has been obtained recently in Ref.[1] where uniform pairings are assumed, $\Delta_i^A = \Delta_A$ for any $i$ in $\mathcal{A}$ and $\Delta_j^B = \Delta_B$ for any $j$ in $\mathcal{B}$. This hypothesis allows great simplifications in the calculations but does not correspond in general (inequivalent orbitals) to the true self-consistent BdG solution. In our general proof, the crucial step is the introduction of a hidden symmetry which splits the BdG eigenstates in two subsets $\mathcal{S}_+$ and $\mathcal{S}_-$. We now show some general properties which result from this. Using Eq.(10), for any $j$ in $\mathcal{B}$, $\Delta_j^B \leq$

$\frac{|U|}{N_c}(\sum_{\mathbf{k},n=1}^{n=\Lambda_B}|b_{nj}^+|^2 + \sum_{\mathbf{k},m=1}^{m=\Lambda_A}|b_{mj}^-|^2) = |U|\langle\hat{n}_{B_j,\uparrow}\rangle = \frac{|U|}{2}$. Similarly, for any $i$, one finds $\Delta_i^A \leq \frac{|U|}{2}$.

If $\langle\Delta^\lambda\rangle$, $\lambda = A, B$, denote the average of the pairings on each sublattice, then,

$$|U|(\langle\Delta^B\rangle - \langle\Delta^A\rangle) = \frac{1}{\Lambda_B}(F_1 - F_2), \qquad (12)$$

where $F_1 = \frac{1}{N_c}\sum_{\mathbf{k},j,n}|b_{nj}^+|^2 + \frac{r}{N_c}\sum_{\mathbf{k},i,n}|a_{ni}^+|^2$ and $F_2 = \frac{1}{N_c}\sum_{\mathbf{k},j,m}|b_{mj}^-|^2 + \frac{r}{N_c}\sum_{\mathbf{k},i,m}|a_{mi}^-|^2$, with $r = \frac{\Lambda_B}{\Lambda_A} \geq 1$. Eigenstates being normalized, implies $F_1 \geq \frac{\Lambda_B}{2}$ and $F_2 \leq \frac{\Lambda_B}{2}$ which demonstrates,

$$\langle\Delta_B\rangle \geq \langle\Delta_A\rangle. \qquad (13)$$

Combining this equation and Eq.(3) gives,

$$\frac{\langle\Delta_B\rangle}{|U|} \geq \frac{r-1}{2r}. \qquad (14)$$

For instance, in the stub lattice ($r = 2$), recently it has been found numerically that the lower bound of $\frac{\langle\Delta_B\rangle}{|U|}$ is 0.25 which coincides exactly with $\frac{r-1}{2r}$ [14].

*Pairings in partially filled flat bands.*-Partially filled FBs for which $\mu = -|U|/2$, correspond to electron density $\nu$ varying between $\nu_{min} = 2\Lambda_A$ and $\nu_{max} = 2\Lambda_B$. For the half-filled case we introduce $\bar{\nu} = \Lambda_A + \Lambda_B$. To calculate the pairings for $\nu_{min} \leq \nu \leq \nu_{max}$, we use the pseudo-spin SU(2) symmetry of the AHM in BLs [4–6], which is a form of rotation invariance in particle-hole space. The AHM is re-expressed,

$$\hat{H} = \sum_{Ii,Jj,\sigma}t_{A_iB_j}^{IJ}\hat{c}_{IA_i,\sigma}^\dagger\hat{c}_{JB_j,\sigma} - \frac{2}{3}|U|\sum_{I,l\lambda=A,B}\hat{\mathbf{T}}_{I\lambda_l}\cdot\hat{\mathbf{T}}_{I\lambda_l}$$
$$- (\mu + |U|/2)\sum_{I,l,\lambda=A,B}\hat{n}_{I\lambda_l}. \qquad (15)$$

The components of the pseudo-spin operator read,

$$\hat{T}_{I\lambda_l}^+ = \eta_\lambda\hat{c}_{I\lambda_l,\uparrow}\hat{c}_{I\lambda_l,\downarrow}, \qquad (16)$$

$$\hat{T}_{I\lambda_l}^- = \eta_\lambda\hat{c}_{I\lambda_l,\downarrow}^\dagger\hat{c}_{I\lambda_l,\uparrow}^\dagger, \qquad (17)$$

$$\hat{T}_{I\lambda_l}^z = \frac{1}{2}(1 - \hat{n}_{I\lambda_l}), \qquad (18)$$

$\eta_\lambda = 1$ (respectively $-1$) if $\lambda = A$ (respectively $B$). These operators obey the usual commutation relations of spin operators. In partially filled FBs, the last term (right side) in Eq.(15) vanishes and, $[\hat{H}, \hat{T}^\pm] = [\hat{H}, \hat{T}^z] = 0$, where $\hat{\mathbf{T}} = \sum_{I,l,\lambda=A,B}\hat{\mathbf{T}}_{I\lambda_l}$ is the total pseudo-spin operator. The Hamiltonian has pseudospin SU(2) symmetry. $\langle\hat{\mathbf{T}}_{I\lambda_l}\rangle_0$ is cell independent and,

$$\langle\hat{\mathbf{T}}_{\lambda_l}\rangle_0 = \begin{bmatrix}\langle\hat{T}_{\lambda_l}^x\rangle_0 = \eta_\lambda\Re(\frac{\Delta_l^\lambda}{|U|}) \\ \langle\hat{T}_{\lambda_l}^y\rangle_0 = \eta_\lambda\Im(\frac{\Delta_l^\lambda}{|U|}) \\ \langle\hat{T}_{\lambda_l}^z\rangle_0 = \frac{1}{2}(1 - n_{\lambda_l})\end{bmatrix}, \qquad (19)$$

$\hat{H}_{BdG}$ is invariant under any identical rotation of the pseudo-spins. We consider $\mathcal{R}_y(\theta)$ the rotation of angle $\theta$ around the $y$-axis,

$$\begin{bmatrix}\hat{c}_{I\lambda_l,\uparrow} \\ \hat{c}_{I\lambda_l,\downarrow}\end{bmatrix} \xRightarrow{\mathcal{R}_y(\theta)} \begin{bmatrix}\cos(\theta/2)\hat{c}_{I\lambda_l,\uparrow} - \eta_\lambda\sin(\theta/2)\hat{c}_{I\lambda_l,\downarrow}^\dagger \\ \cos(\theta/2)\hat{c}_{I\lambda_l,\downarrow} + \eta_\lambda\sin(\theta/2)\hat{c}_{I\lambda_l,\uparrow}^\dagger\end{bmatrix}. (20)$$

Let us assume that the self-consistent solution for $\nu = \bar{\nu}$ is known. The expectation value of the corresponding pseudo-spins reads,

$$\bar{\mathbf{T}}_{\lambda_l} = \begin{bmatrix}\bar{T}_{\lambda_l}^x = \eta_\lambda\frac{\bar{\Delta}_l^\lambda}{|U|} \\ \bar{T}_{\lambda_l}^y = 0 \\ \bar{T}_{\lambda_l}^z = 0\end{bmatrix}. \qquad (21)$$

$\bar{T}_{\lambda_l}^y$ and $\bar{T}_{\lambda_l}^z$ vanish since (i) the pairings are taken real and (ii) because of the uniform density theorem [12]. Applying $\mathcal{R}_y(\theta)$ to the pseudo-spins leads to a BdG solution corresponding to a partial filling of the FBs,

$$\Delta_l^\lambda = \bar{\Delta}_l^\lambda\cos(\theta), \qquad (22)$$

$$n_{\lambda_l} = 1 + 2\eta_\lambda\frac{\bar{\Delta}_l^\lambda}{|U|}\sin(\theta). \qquad (23)$$

The corresponding filling factor is,

$$\nu(\theta) = \bar{\nu} + \sin(\theta)(\Lambda_B - \Lambda_A). \qquad (24)$$

We emphasize that Eq.(3) has been used. Hence, $\theta = \pi/2$ corresponds to the fully filled FBs, i.e. $\nu = \nu_{max}$ while $\theta = -\pi/2$ to empty FBs or $\nu = \nu_{min}$. Combining Eq.(22) and Eq.(24), one obtains,

$$\Delta_l^\lambda = \bar{\Delta}_l^\lambda f(\nu), \qquad (25)$$

$$n_{\lambda_l} = 1 + 2\eta_\lambda\frac{\bar{\Delta}_l^\lambda}{|U|}\sqrt{1 - f^2(\nu)}, \qquad (26)$$

where,

$$f(\nu) = \frac{2}{\nu_{max} - \nu_{min}}\sqrt{(\nu - \nu_{min})(\nu_{max} - \nu)}. \quad (27)$$

Similar expressions have been derived in Ref.[1], where a uniform pairing is forced on the orbitals on the dominant lattice. Our proof is general, without restriction on the pairings, and requires only that the sum-rule given in Eq.(3) has been proved.

*The superfluid weight in partially filled FBs.*- Here, we derive a general relationship between $D^s$ in partially filled FBs and that of half-filled BL. The SFW is defined as [2, 3],

$$D_\mu^s = \frac{1}{N_c}\frac{\partial^2\Omega(\mathbf{q})}{\partial q_\mu^2}\Big|_{\mathbf{q}=\mathbf{0}}, \qquad (28)$$

$\Omega(\mathbf{q})$ is the grand-potential and $q$ mimics the effect of a vector potential, introduced by a standard Peierls substitution.

Recently, it has been argued that when the quantum metric (QM) [20, 21] associated to FBs is not minimal, corrections should be included in Eq.(6) [1]. Contrary to $D_\mu^s$, the QM which measures the typical spreading of the FB eigenstates is a quantity which depends on the orbital positions. However, for any BL, one can always find the orbital positions which minimize the QM, therefore, for which Eq.(6) is correct. It generally corresponds to the most symmetrical positions of the orbitals within the cell. Following Refs.[22] and [23] leads to,

$$D_\mu^s = \frac{2}{N_c} \sum_{\mathbf{k},mn} \frac{J_\mu^{nm}}{E_n^< - E_m^>}, \qquad (29)$$

where $J_\mu^{nm} = |\langle \Psi_n^< | \hat{V}_\mu | \Psi_m^> \rangle|^2 - |\langle \Psi_n^< | \hat{\Gamma} \hat{V}_\mu | \Psi_m^> \rangle|^2$, with $\hat{\Gamma} = \mathrm{diag}(\hat{\mathbb{1}}_{\Lambda\times\Lambda}, -\hat{\mathbb{1}}_{\Lambda\times\Lambda})$ and $\hat{V} = \mathrm{diag}(\hat{v}^0, \hat{v}^0)$. The velocity operator along the $\mu$-direction is $\hat{v}_\mu^0 = \frac{\partial \hat{h}^0}{\partial k_\mu}$ where $\hat{h}^0 = \begin{bmatrix} 0 & \hat{h}_{AB} \\ \hat{h}_{AB}^\dagger & 0 \end{bmatrix}$.

To avoid confusion due to multiple indices, we introduce here the notation $|\Psi_m^>\rangle = (|a_m^>\rangle, |b_m^>\rangle, |\bar{a}_m^>\rangle, |\bar{b}_m^>\rangle)^t$ for the eigenstates with positive energy $E_m^>$, similarly $|\Psi_n^<\rangle = (|a_n^<\rangle, |b_n^<\rangle, |\bar{a}_n^<\rangle, |\bar{b}_n^<\rangle)^t$ for those with negative energy $E_n^<$. Thus, we ignore whether these states belong to $\mathcal{S}^\pm$. The eigenstates for $\nu = \bar{\nu}$ are specified by simply replacing $n \to 0n$ and $m \to 0m$.

Assuming the eigenstates known for $\nu = \bar{\nu}$, $D_\mu^s$ in partially filled FBs is obtained using the pseudospin SU(2) symmetry of the Hamiltonian. Recall that the quasi particle eigenvalues are invariant under the pseudospin rotation. From Eq.(20) the rotated eigenstates are, $|\psi_n^<\rangle = \hat{U}_\theta |\Psi_{0n}^<\rangle$ (similarly $|\Psi_m^>\rangle = \hat{U}_\theta |\Psi_{0m}^>\rangle$) where,

$$\hat{U}_\theta = \begin{bmatrix} c & 0 & s & 0 \\ 0 & c & 0 & -s \\ -s & 0 & c & 0 \\ 0 & s & 0 & c \end{bmatrix}, \qquad (30)$$

with $c = \cos(\theta/2)$ and $s = \sin(\theta/2)$. The matrix elements in Eq.(29) are given by,

$$\langle \Psi_n^< | \hat{\Gamma}^p \hat{V}_\mu | \Psi_m^> \rangle = \langle \Psi_{0n}^< | \hat{U}_{-\theta} \hat{\Gamma}^p \hat{V}_\mu \hat{U}_\theta | \Psi_{0m}^> \rangle, \quad (31)$$

where $p = 0$ or $1$. $|\bar{a}_{0n}^<\rangle = \epsilon_n |a_{0n}^<\rangle$ and $|\bar{b}_{0n}^<\rangle = -\epsilon_n |b_{0n}^<\rangle$ where $\epsilon_n = 1$ (respectively $-1$) if $|\Psi_{0n}^<\rangle \in \mathcal{S}^+$ (respectively $\in \mathcal{S}^-$). We proceed similarly with $|\Psi_{0m}^>\rangle$ and get,

$$J_\mu^{nm} = |C_{0nm}^{<,>}|^2 g_{nm}, \qquad (32)$$

where $g_{nm} = ((1-\epsilon_n\epsilon_m)c + (\epsilon_n+\epsilon_m)s)^2 - (1+\epsilon_n\epsilon_m)^2$ and $C_{0nm}^{<,>} = \langle a_{0n}^< | \partial_\mu \hat{h}_{AB} | b_{0m}^> \rangle + \langle b_{0n}^< | \partial_\mu \hat{h}_{AB}^\dagger | a_{0m}^> \rangle$. Eq.(32) can be simplified and gives,

$$J_\mu^{nm} = -4\epsilon_n\epsilon_m |C_{0nm}^{<,>}|^2 \cos^2(\theta). \qquad (33)$$

Using Eq.(24), we then end up with,

$$D_\mu^s(\nu) = f^2(\nu) D_\mu^s(\bar{\nu}). \qquad (34)$$

In partially filled FBs, $D_\mu^s$ always has a universal parabolic shape and vanishes for $\nu = \nu_{min}$ and $\nu_{max}$. To derive Eq.(7), one needs Eq.(3). Note that Eq.(33) indicates that the contributions to $D_\mu^s(\nu)$ originating from pairs of eigenstates in the same subspace $\mathcal{S}^+$ or $\mathcal{S}^-$ are positive, while they are negative in the other case.

*Effects of disorder*.- We have previously considered clean systems. An interesting question is: What is the impact of disorder that preserves the bipartite character of the lattice such as random hoppings or introduction of vacancies? Translation invariance being broken, $\hat{H}_{BdG}$ must be diagonalized in real space. The number of zero energy eigenstates is $\mathcal{N}_{E=0} = |\mathcal{N}_\mathcal{B} - \mathcal{N}_\mathcal{A}|$ where $\mathcal{N}_\lambda$ is the total number of orbitals $\lambda = \mathcal{A}, \mathcal{B}$. In the clean case, our proofs are based on Lemma 1 and Lemma 2, which remain valid in the single cell made up of $\mathcal{N}_\mathcal{A}$ A-orbitals and $\mathcal{N}_\mathcal{B}$ B-orbitals. Thus, in the disordered half-filled BL, Eq.(3) becomes,

$$\sum_{j=1}^{\mathcal{N}_\mathcal{B}} \Delta_j^B - \sum_{i=1}^{\mathcal{N}_\mathcal{A}} \Delta_i^A = \frac{|U|}{2} |\mathcal{N}_\mathcal{B} - \mathcal{N}_\mathcal{A}|, \qquad (35)$$

where $i$ (respectively $j$) runs now over the whole sublattice $\mathcal{A}$ (respectively $\mathcal{B}$). In addition, Eq.(25) and Eq.(7) which give the filling dependence of the pairings and the SFW are valid as well.

Notice that BCS wavefunction being the exact groundstate in BL hosting isolated FBs when $|U|$ is smaller than the gap [24], implies as well the exactness of our results in this limit. Thus, it would be of great interest to confirm this statement from exact methods such as DMRG, a reliable and well suited tool for quasi one-dimensional systems.

To conclude, using a hidden symmetry of the BdG eigenstates, we have rigourously demonstrated that in bipartite lattices the pairings and the SFW obey universal relations. Furthermore, these general properties are shown to hold in disordered systems as long as the bipartite character of the lattice is conserved. Our findings could have an important impact in the search of novel families of compounds exhibiting unconventional FB superconductivity.

* E-mail:georges.bouzerar@neel.cnrs.fr

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

## SUPPLEMENTAL MATERIAL

The purpose of this supplemental material is to illustrate the sum-rules and other relations demonstrated in the general context of bipartite lattices (BLs) where flat bands (FBs) are either half-filled or partially filled. The prototype of two dimensional BL considered here, we will designate it by $\mathcal{L}$-lattice, is shown in Fig.1. The $\mathcal{L}$-lattice consists in two sublattices $\mathcal{A}$ and $\mathcal{B}$ which contain respectively $\Lambda_A = 3$ and $\Lambda_B = 5$ orbitals per unit cell, where $\mathcal{A} = \{A_1, A_2, A_3\}$ and $\mathcal{B} = \{B_1, B_2, ..., B_5\}$ . In the absence of electron-electron interaction, the one-particle spectrum consists exactly in $N_{fb} = \Lambda_B - \Lambda_A = 2$ FBs with energy $E_{fb} = 0$ and $2\Lambda_A = 6$ symmetric dispersives bands (chirality).

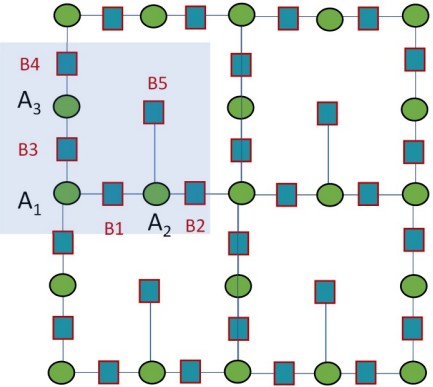

Figure 1. (Color online) Prototype of two-dimensional bipartite lattice ($\mathcal{L}$), with $\Lambda_A = 3$ atoms of type A and $\Lambda_B = 5$ atoms of type B per unit cell (shaded area). The hoppings are restricted to nearest neighbors only, they are all equal, set to 1. The single particle Hamiltonian has two degenerate flat bands.

### Symmetry in the $H_{BdG}$ eigenstates

It has been shown in the main text, that eigenstates of the Bogoliubov de Gennes Hamiltonian $H_{BdG}$ as given in Eq.(2) in the article, can be divided into two subsets $\mathcal{S}_+$ and $\mathcal{S}_-$ which are defined in what follows. Consider a normalized eigenstate of $H_{BdG}$, $|\Psi\rangle = (|u\rangle, |v\rangle)^t$ of energy $E$, where $|u\rangle = (|\mathbf{a}\rangle, |\mathbf{b}\rangle)^t$ and $|v\rangle = (|\bar{\mathbf{a}}\rangle, |\bar{\mathbf{b}}\rangle)^t$. The columns $|\mathbf{a}\rangle$ and $|\bar{\mathbf{a}}\rangle$ (respectively $|\mathbf{b}\rangle$ and $|\bar{\mathbf{b}}\rangle$) are of length $\Lambda_A$ (respectively $\Lambda_B$),

$$|\Psi\rangle \in \mathcal{S}_+ \Leftrightarrow |\bar{\mathbf{a}}\rangle = |\mathbf{a}\rangle, |\bar{\mathbf{b}}\rangle = -|\mathbf{b}\rangle, \qquad (1)$$

$$|\Psi\rangle \in \mathcal{S}_- \Leftrightarrow |\bar{\mathbf{a}}\rangle = -|\mathbf{a}\rangle, |\bar{\mathbf{b}}\rangle = |\mathbf{b}\rangle. \qquad (2)$$

For a given value of the electron-electron interaction $|U|$, here we have chosen $|U| = 3$, Fig.2 depicts the QP dispersions with negative energy in the half-filled $\mathcal{L}$−lattice and along the $\Gamma M$ direction in the Brillouin zone. Unambiguously, for any value of the momentum $\mathbf{k}$, the spec-

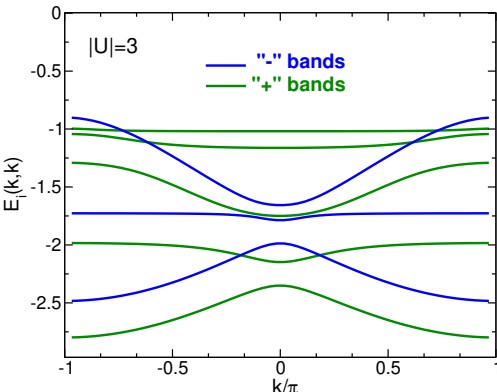

Figure 2. (Color online) Negative part of the quasiparticle dispersions in the $(1,1)$-direction for the half-filled BL $\mathcal{L}$ as depicted in Fig.1. The green (respectively blue) line corresponds to QP eigenstates in $\mathcal{S}_+$ (respectively $\mathcal{S}_-$). There are $\Lambda_A = 3$ bands in $\mathcal{S}_-$ and $\Lambda_B = 5$ in $\mathcal{S}_+$. Here, the on-site interaction parameter $|U| = 3$, the conclusion is the same for any $|U|$.

trum consists in $\Lambda_A = 3$ eigenstates in $\mathcal{S}_-$ and $\Lambda_B = 5$ eigenstates in $\mathcal{S}_+$.

## Sum rule for the pairings in half-filled bipartite lattices

In the main text of the article we it has been rigourously proved that, in any half-filled bipartite lattice the pairings obey the following sum-rule,

$$\sum_{j=1}^{\Lambda_B} \Delta_j^B - \sum_{i=1}^{\Lambda_A} \Delta_i^A = \frac{|U|}{2}(\Lambda_B - \Lambda_A). \tag{3}$$

As an illustration, in Fig.3, the pairings for each orbital are plotted as a function of $|U|$ in the half-filled $\mathcal{L}$-lattice. For obvious symmetry reasons (see Fig.1), one finds that $\Delta_1^B = \Delta_2^B$ and $\Delta_3^B = \Delta_4^B$. As it can be clearly seen, for any $|U|$, Eq.(3) is exactly fulfilled.

If we define the average value of the pairing on both sublattices by $\langle \Delta_\lambda \rangle$ where $\lambda = A, B$. For any $|U|$, we have shown in the main text that,

$$\langle \Delta_B \rangle \geq \langle \Delta_A \rangle, \tag{4}$$

and found as well a lower bound for the average value of the pairings on $\mathcal{B}$-sublattice,

$$\frac{\langle \Delta_B \rangle}{|U|} \geq \frac{r-1}{2r}, \tag{5}$$

where $r = \Lambda_B / \Lambda_A$ has been introduced.

In the case of the half-filled $\mathcal{L}$-lattice, Fig.3 clearly shows that $\langle \Delta_B \rangle \geq \langle \Delta_A \rangle$ for any $|U|$. According to Eq.(5), in the present case, one expects that $\langle \Delta_B \rangle \geq 0.2\,|U|$. This is indeed in perfect agreement with the results depicted in Fig.3. More precisely, the lower bound is found to coincide exactly with $\frac{\partial \langle \Delta_B \rangle}{\partial |U|}\Big|_{U=0}$.

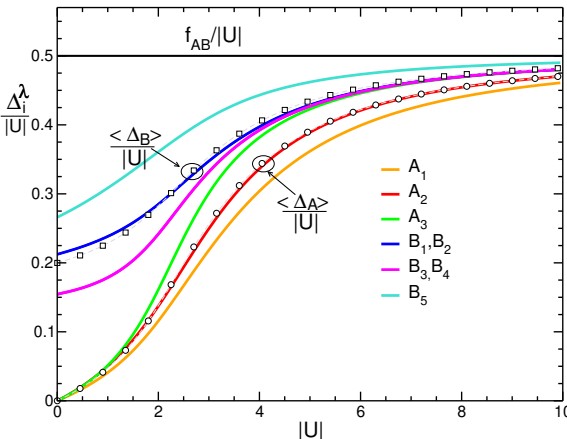

Figure 3. (Color online) Pairings (divided by $|U|$) as a function of $|U|$ for the half-filled BL depicted in Fig.1. The open circles (respectively squares) are the average values of the pairings on sublattice $\mathcal{A}$ (respectively $\mathcal{B}$). The horizontal black line corresponds to $f_{AB}/|U|$ where $f_{AB} = \frac{1}{2}(\sum_{j=1}^{\Lambda_B} \Delta_j^B - \sum_{i=1}^{\Lambda_A} \Delta_i^A)$.

## The superfluid weight in partially filled FBs

In the main text we have proved a general relationship between the superfluid weight (SFW) $D_\mu^s$ in partially filled FBs and that of the half-filled lattice. The SFW is defined as [2, 3],

$$D_\mu^s = \frac{1}{N_c} \frac{\partial^2 \Omega(\mathbf{q})}{\partial q_\mu^2}\Big|_{\mathbf{q}=\mathbf{0}}, \tag{6}$$

$\Omega(\mathbf{q})$ is the grand-potential and $q$ mimics the effect of a vector potential, introduced by a standard Peierls substitution in the hopping terms in the BdG Hamiltonian.

First, we have carefully checked in the case of the $\mathcal{L}$-lattice that (i) the corrections to Eq .(6) as discussed in Ref.[1] are vanishing and (ii) the quantum metric associated to the FBs is minimal for the geometry depicted in Fig.1. Using the pseudo-spin SU(2) symmetry of the Hamiltonian for $\mu = -|U|/2$ [4–6], it has been shown in the main text that,

$$D_\mu^s(\nu) = f^2(\nu) D_\mu^s(\bar{\nu}), \tag{7}$$

where the filling dependent function is,

$$f(\nu) = \frac{2}{\nu_{max} - \nu_{min}} \sqrt{(\nu - \nu_{min})(\nu_{max} - \nu)}. \tag{8}$$

Thus, the SFW for partially filled FBs always has a universal parabolic shape and $D_\mu^s(\nu)$ vanishes for $\nu = \nu_{min}$ and $\nu = \nu_{max}$. These fillings correspond respectively to empty FBs for which $\nu = \nu_{min} = 2\Lambda_A = 6$ and fully filled FBs where $\nu = \nu_{max} = 2\Lambda_B = 10$. As an illustration,

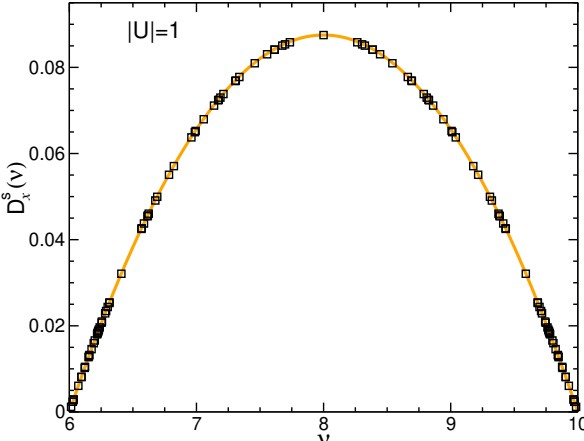

Figure 4. (Color online)Superfluid weight $D_x^s$ as a function of the electron filling in the $\mathcal{L}$-lattice. The densities correspond to partially filled FBs where the chemical potential is $\mu = -|U|/2$. The electron interaction parameter is $|U| = 1$. The symbols are the numerical data and the continuous line is the analytical expression as discussed in the main text and given by Eq.(7).

Fig.4 depicts $D_x^s(\nu)$ as a function of $\nu$ in the $\mathcal{L}$-lattice. As it is clearly seen, the agreement between the numerical data and the analytical expression given in Eq.(7) is excellent.

## The impact of the disorder: the case of randomly distributed vacancies.

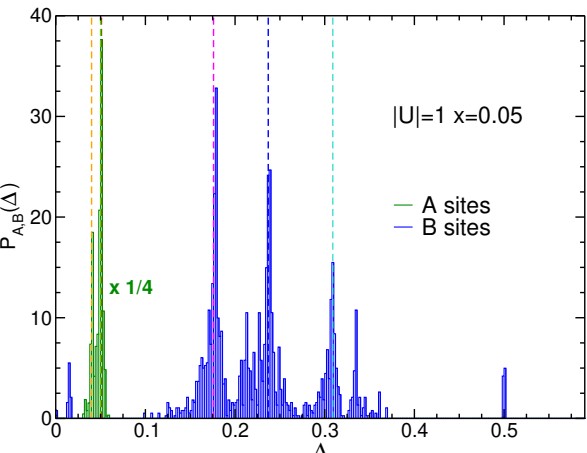

Figure 5. (Color online) Distribution of the pairings $P_\Lambda(\Delta)$ in both sublattices ($\Lambda = A, B$) for the disordered $\mathcal{L}$-lattice. The concentration of randomly distributed vacancies, $x$ is the ratio between the number of vacancies and that of orbitals in the pristine system which here contains $\mathcal{N} = 8 \times 20^2$ orbitals. The system is half-filled and $|U| = 1$. For more visibility the probability distribution $P_A(\Delta)$ has been multiplied by 1/4. The vertical dashed lines are the value of the pairings in the clean system.

In our manuscript, it has been shown that in the case of disorder that conserves the bipartite character of the lattice the sum-rules and other relations established in the case of clean systems still hold. Here, our puropose is to illustrate this feature. We consider the impact of vacancies randomly distributed in the $\mathcal{L}-$lattice. In the case of disordered half-filled system, it has been argued in the main text that Eq.(3) becomes,

$$\sum_{j=1}^{\mathcal{N}_\mathcal{B}} \Delta_j^B - \sum_{i=1}^{\mathcal{N}_\mathcal{A}} \Delta_i^A = \frac{|U|}{2}|\mathcal{N}_\mathcal{B} - \mathcal{N}_\mathcal{A}|, \qquad (9)$$

where $i$ (respectively $j$) runs now over the whole sublattice $\mathcal{A}$ (respectively $\mathcal{B}$), and $\mathcal{N}_\mathcal{A}$ (resp. $\mathcal{N}_\mathcal{B}$) are the total number of A-orbitals (respectively B-orbitals) in the disordered lattice.

Because of the loss of translation invariance, the calculations require multiple real space diagonalizations of the BdG Hamiltonian, until convergence in the self-consistent loop is reached. The size of the matrices is $2\mathcal{N} \times 2\mathcal{N}$ where $\mathcal{N} = \mathcal{N}_\mathcal{A} + \mathcal{N}_\mathcal{B}$. For our illustration, we have considered a system that contains about 3200 orbitals. In Fig.5, the pairing distribution in the disordered half-filled $\mathcal{L}$-lattice is depicted. The configuration of disorder corresponds to the introduction of 5% of vacancies randomly distributed. We have checked that Eq.(9) is exactly verified, as well as the relation $\langle \Delta_B \rangle \geq \langle \Delta_A \rangle$ which could be anticipated from the plot of the pairing distributions. Additionally, we have checked that Eq.(5) is as well fulfilled $\frac{\langle \Delta_B \rangle}{|U|} \geq \frac{1}{2}(1 - \frac{1}{r})$, where in the disordered lattice $r = \frac{\mathcal{N}_\mathcal{B}}{\mathcal{N}_\mathcal{A}}$.

---

* E-mail:georges.bouzerar@neel.cnrs.fr
[1] K.-E. Huhtinen, J. Herzog-Arbeitman, A. Chew, B. A. Bernevig, and P. Törmä, Phys. Rev. B **106**, 014518 (2022).
[2] B. S. Shastry and B. Sutherland, Phys. Rev. Lett. **65**,243 (1990).
[3] D. J. Scalapino, S. R. White, and S. C. Zhang, Phys. Rev. Lett. **68**, 2830 (1992); Phys. Rev. B **47**, 7995 (1993).
[4] C. N. Yang, Phys. Rev. Lett. **63**, 2144 (1989)
[5] S. Zhang, Phys. Rev. Lett. **65**, 120 (1990).
[6] M. Mermele, Phys. Rev. B **76**, 035125 (2007).