# Peer review of "Universal relations in flat band superconducting bipartite lattices"

_SciPost Physics_

## Round 1 · Referee Report · Anonymous · 2023-11-5

Report

This manuscript studies some aspects of superconductivity, specifically various "universal" relations between the pairing amplitudes and superfluid stiffness, in “flat-band” systems in the presence of *only* an on-site attractive-U Hubbard interaction, and assuming (without any justification; see below) that mean-field theory is applicable. Unfortunately, the paper does make some misleading statements (perhaps unintentionally), especially when it comes to presenting results using the said mean-field theory as if they hold well beyond this uncontrolled and unjustified limit. At the same time, the paper fails to cite many important papers, including an early paper by Tovmasyan et al, Phys. Rev. B 94, 245149 (2016), which beautifully highlighted many important aspects of the flat-band problem with on-site attractive-U Hubbard interaction, including where BCS theory is valid. As shown by these authors, the only limit where the BCS wavefunction is the actual ground-state wavefunction, thereby providing some a posteriori justification, is tied to an emergent SU(2) symmetry in the problem. Furthermore, they also derived relations between the superfluid stiffness and pairing amplitudes (they are both related to the interaction strength, as well as some other invariants). However, in this limit even though the stiffness is finite, the transition temperature vanishes.

I do not find this manuscript well written/presented and contributing any fundamental new insights into the problem (though it tries to present results in a “rigorous” fashion, even when the starting point of the approach lacks much rigor). Ultimately, all of the calculations involve various manipulations of the same BdG equations. At this point, based on the large existing literature of similar analysis of the FB problem using mean-field theory, I find that this work is just a straightforward extension of these ideas and does not meet the criteria for acceptance in SciPost.

Interestingly, there is repeated mention of how well BCS mean-field theory apparently agrees with DMRG. However, as has been pointed out in various unbiased, large-scale two-dimensional QMC simulations, BCS mean-field theory also fails in dramatic ways once the on-site attraction is perturbed by infinitesimal nearest-neighbor interactions (Hofmann et al., ‘20, ‘23; Peri et al., ‘21). The authors do not cite any of these papers as well.

---

## Round 1 · Referee Report · Anonymous · 2023-12-11

Strengths

Presents a detailed and explicit solution of a mean-field model.

Weaknesses

1. Validity beyond mean-field is not clear or even properly addressed.
2. Lots of relevant work is not cited
3. Broad applicability is unclear.

Report

The manuscript introduces and solves a model for BCS pairing in the attractive Hubbard model on bipartite lattices. The bipartite structure introduces a (well-known) pseudospin SU(2) symmetry that allows some additional analytical tractability. The authors prove various results about this model within the framework of the Bogoliubov-de Gennes (BdG) mean field equations, and use this to derive bounds on the superfluid stiffness etc. The key step is the identification of a "hidden symmetry" in the BdG eigenstates, i.e. a property of the mean-field solutions.

I am not persuaded of the validity of the results beyond the mean-field limit assumed without any clear justification. Indeed, there are convincing results (see e.g. PRB 94, 245149) suggesting that the exact BCS wavefunction and hence the BdG solutions are only valid in some special limits. Furthermore given that bipartite lattices often admit sign-free Quantum Monte Carlo there exist unbiased numerical studies of the attractive-U Hubbard problem on such lattices even with flat bands (e.g. PRB 102, 201112), but no effort is made by the present authors to compare to these results or address their implications for their work. This lack of any embedding of their work into the very intense and broader range of activity in the field to my mind actually raises the issue to one of poor writing, and as such it is very difficult to assess the significance or originality of the work (which is not to immediately say that it is unoriginal or insignificant, only that a reader is forced to suspend their judgement in the absence of evidence.)

At present I believe the manuscript falls far below the criteria for acceptance in SciPost Physics. At minimum, the authors must explain the extent to which their results are valid outside the mean-field model, or else reframe the paper as a series of exact results on the mean field model, in order to be able to make a fair assessment on whether the paper may be suitable. I believe there may exist some argument to be made that new exact results on a mean field model could be of interest but the bar is high, and if there is not a plausible case to be made of either the importance of these new results or for their validity beyond mean field my recommendation would be to reject; however I am willing to consider that authors may be able to rebut these criticisms in a major revision.

Requested changes

See report.

---

## Round 1 · Referee Report · Anonymous · 2023-12-18

Strengths

1 - Extensions of results for flat-band superconductivity beyond the uniform pairing condition
2 - Careful mean-field study of BdG Hamiltonians on biparitite lattices

Weaknesses

1 - No discussion of beyond mean-field effects
2 - Lack of clear discussion of the novelty of the findings

Report

The authors study the solution of the Bogoliubov- de Gennes Hamiltonian on bipartite lattices. They focus on the zero-temperature limit and impose time-reversal symmetry. They find bounds on the pairing strength in terms of the ratio between the number of lattice sites in each sublattice and how the superfluid weight depends on the filling of the flat band.

Their findings agree with previous results reported in the literature and are not entirely novel. The main advancement is the identification of a symmetry in the BdG equations on bipartite lattices. This symmetry allows them to extend previous results beyond the uniform pairing condition, i.e., the assumption of equal pairing on all sublattices on which the flat band eigenstates have non-zero weight.

What is lacking and necessary is a better framing in the existing literature. In particular, the authors should better address the regime of validity of their findings. PRB 94, 245149, and Nat. Comm. 6, 8944 show how the BCS wavefunction is an exact zero-temperature ground state of isolated flat bands with on-site attractive interactions. While the latter is restricted to bipartite lattices, the first extends this result beyond bipartite lattices with uniform pairing. Further evidence for the validity of the BCS approximation at zero temperature is provided by the quantum Monte Carlo (QMC) results of PRB 102, 201112; PRL 130, 226001; PRL 128, 087002; PRL 126, 027002.
The authors could use the above references to better argue for the validity of their mean-field approach. For example, the SU(2) symmetry they find at the mean-field level, would at face value forbid any finite-temperature transition in 2D. The QMC results show how this problem is circumvented beyond mean-field.

The authors should address at least the following questions:

- What happens without time-reversal symmetry? Which result will break down?
- What will happen in the presence of intra-sublattice pairing or longer-range interactions?
- They should distinct between gapless and gapped flat bands. In the latter, the mean-field treatment seems better justified.
- Which results are strictly valid for the flat bands and which for any band of the bipartite lattice?

Lastly, to meet the criteria upheld by sciPost, the authors should specify the relevance of their findings. I think they could stress more the importance of extending previous results beyond the uniform pairing condition.

Some minor remarks on the formatting of the paper:
- The authors should present the non-interacting band structure of the tight-binding model studied in the supplemental material.
- There seems to be an issue with the citation of equations and references. For example, below Eq. (35) the authors refer to Eq. (7) of the supplemental material rather than Eq. (34) of the main text. Similarly, below Eq. (11) they cite Ref. (1) rather than Ref. 13.
- Ref. 9 is repeated in Ref. 10.

---

## Editorial Decision

resubmitted